# A Novel Study on a Generalized Model Based on Self-Supervised Learning and Sparse Filtering for Intelligent Bearing Fault Diagnosis

**DOI:** 10.3390/s23041858

**Published:** 2023-02-07

**Authors:** Guocai Nie, Zhongwei Zhang, Mingyu Shao, Zonghao Jiao, Youjia Li, Lei Li

**Affiliations:** School of Transportation and Vehicle Engineering, Shandong University of Technology, Zibo 255000, China

**Keywords:** self-supervised learning, sparse filtering, bearing fault diagnosis, deep learning

## Abstract

Recently, deep learning has become more and more extensive in the field of fault diagnosis. However, most deep learning methods rely on large amounts of labeled data to train the model, which leads to their poor generalized ability in the application of different scenarios. To overcome this deficiency, this paper proposes a novel generalized model based on self-supervised learning and sparse filtering (GSLSF). The proposed method includes two stages. Firstly (1), considering the representation of samples on fault and working condition information, designing self-supervised learning pretext tasks and pseudo-labels, and establishing a pre-trained model based on sparse filtering. Secondly (2), a knowledge transfer mechanism from the pre-training model to the target task is established, the fault features of the deep representation are extracted based on the sparse filtering model, and softmax regression is applied to distinguish the type of failure. This method can observably enhance the model’s diagnostic performance and generalization ability with limited training data. The validity of the method is proved by the fault diagnosis results of two bearing datasets.

## 1. Introduction

With the continuous improvement of science and technology, high-end mechanical equipment tends to be characterized by high speed, high power, and high precision, which causes the fault diagnosis of this type of equipment to face great challenges [1,2]. As important components of modern mechanical equipment, bearings are widely used in production and life [3,4], for instance, in automobiles, ships, airplanes, generators, and household appliances, as well as in mining plants, electrical plants, printing plants, textile plants, and medical equipment. The normal operation of the bearing can guarantee the smooth running of the mechanical equipment, can guarantee the economy of the enterprise, and can improve the efficiency of production. However, during the long-term operation of bearings, there may be rolling element failures, inner ring failures, outer ring failures, etc., which have a great impact on the benefits to the company and which even threaten personal safety in severe cases [5,6,7,8]. Therefore, in order to guarantee the normal operation of the mechanical equipment, there are important security for condition monitoring and fault diagnosis of the bearings [9].

In general, the learning process of the most intelligent fault diagnosis work is divided into two steps: feature extraction and feature classification [10]. However, the collected vibration signals may have heavy noise, so it is necessary to use advanced signal processing technology for feature extraction of weak fault signals [11]. In recent years, many advanced signal processing techniques, such as time–domain processing [12], frequency–domain processing, and time–frequency transformation [13], have been successfully applied in signal-processing. Gao et al. used time–domain, frequency–domain, and time–frequency–domain analysis to preprocess the collected original gear box data and extract representative features. Finally, a support vector machine is used to classify faults [14]. After fault features are extracted from the original data, Support Vector Machines (SVM) [15,16], K-Nearest Neighbors (KNN) [17], and softmax regression can be used for fault classification.

It can be found that intelligent fault diagnosis research has achieved many good results [18]. However, the traditional intelligent fault diagnosis method also has some disadvantages. Much depends on the expertise and experience of the researcher in the diagnostic process. Second, the model may require redesigning when faced with new diagnostic problems. To overcome these shortcomings, deep learning has attracted more and more attention because it can automatically extract sample features from complex data and apply nonlinear multi-hidden layer networks to express complex feature information. In addition, deep learning can efficiently process a large number of data signals and provide accurate classification results, so it may be a promising tool for processing mechanical big data [19,20,21,22].

Currently, a variety of intelligent deep learning methods such as GAN (generative adversarial network) [23], Stacked Autoencoder (SAE) [24], convolutional neural networks (CNN) [25], and Restricted Boltzmann Machine (RBM) [26], etc. have been successfully applied in many fields. Xu et al. combined the improved GAN with Continuous Wavelet Transform (CWT) and applied it to the field of fault diagnosis. CWT solves the problems of too few samples and sample annotations and effectively enhances the veracity of fault diagnosis [27]. Aiming to address the fault diagnosis problem of rolling bearings, Du et al. established a fault diagnosis approach based on optimized and improved stacked denoising autoencoders, which effectively improved the extraction ability of the fault features [28]. Zhang et al. established a multi-modal attention convolutional neural network to effectively boost the precision of fault diagnosis, aiming at the problem of poor fault diagnosis due to the changing working environment and insufficient samples of rolling bearings [29]. In view of the problem that traditional fault diagnosis methods rely on artificial feature extraction and diagnosis expertise, Tang et al. proposed a deep convolutional network fault diagnosis method, and the experiment proved that the method has good generalization performance [30].

Although deep learning has made massive strides in the field of mechanical fault diagnosis, there is a common disadvantage for most deep learning methods: excessive reliance on a large amount of labeled training data and poor model generalization ability. In actual working conditions, when the bearing fails, the mechanical equipment cannot run for a long period of time, so it is very difficult for us to obtain extensive marked failure data [31,32]. When only a small amount of labeled data is available, it is difficult for deep learning to perform a good fault diagnosis task. How to use a small amount of labeled data to improve the feature extraction ability of the model has become the main topic of current research [33]. Researchers try to solve this problem using semi-supervised learning [34,35] and transfer learning [36]. L.A. Bull et al. proposed a semi-supervised Gaussian mixture model for the signal label classification problem, which obviously enhances the classification accuracy of the model and does not require further inspection of the system [34]. Wu et al. combined semi-supervised learning with a hybrid classification autoencoder and used a softmax classifier to classify fault features [35]. Guo et al. established a multi-task CNN based on knowledge transfer, which dramatically enhances the efficiency and precision of the model [36]. These methods have achieved certain results in fault diagnosis with limited training data, but the working performance does not satisfy the demands of actual working conditions. Compared with semi-supervised learning methods, self-supervised learning methods can facilitate learning useful information in large-scale datasets without any manual annotation, and such methods have better generalization ability.

Through the above discussion, this paper proposes a new intelligent fault diagnosis framework integrating self-supervised learning and a sparse filtering model and applies it to the fault identification of bearings. Firstly, the pretext tasks and corresponding pseudo-labels that can represent sample fault information and working condition information are set up to complete the pre-training process of the sparse filtering model, and the signal transformation types are identified in the pretext tasks. Then, by establishing a parameter transfer mechanism, the valuable information learned by the model is transferred to the target task. Finally, the bearing fault type identification is realized through the softmax regression.

The main research and conclusions of this paper are described as follows:

(1) In this paper, a new intelligent fault diagnosis framework is proposed to achieve the efficient diagnosis of bearing faults with minimal training samples.

(2) Based on the representation of samples for the information of the fault and working condition, self-supervised learning pretext tasks and pseudo-labels are designed. The effective information extracted by pre-training is applied to the sparse filtering feature extraction through the knowledge transfer mechanism, which improves the generalization performance of the model.

(3) The diagnostic results of two bearing fault diagnosis experiments prove the effectiveness and generalization ability of GSLSF in the case of minimal training data.

The structure of this paper is described as follows. Section 2 introduces softmax regression and self-supervised learning. The intelligent fault diagnosis model framework is described in Section 3. Section 4 tests the proposed method experimentally. Section 5 summarizes the research content of this paper.

## 2. Related Work

At present, self-supervised learning is becoming more and more popular because it can reduce the cost of annotating large-scale datasets. It can use custom pretext tasks and pseudo-labels for multiple downstream tasks [37]. This section mainly introduces the latest progress of self-supervised learning.

Self-supervised learning has made great progress in the field of image processing, for example, in image rotation [38], image completion [39,40], image coloring [41,42], image super-resolution [43], image clustering [44,45], etc. Because the mechanical vibration signal is a strong periodic non-stationary signal, and its characteristics are completely different from the image data, it is difficult to design an appropriate agent task for the mechanical vibration signal using the image processing method. Therefore, according to the characteristics of mechanical vibration signals, it is worthwhile studying these signals to find a self-monitoring method. Previously, in the field of fault diagnosis, scholars have conducted little research on self-supervised learning methods. For example, Senanayaka et al. combined support vector machine (SVM) and convolutional neural network (CNN) to realize online detection of multiple faults and fault levels under different speeds and loads. In this method, the self-supervised learning of the proposed CNN algorithm allows online diagnosis based on the latest data learning features, and the effectiveness of this scheme is verified by experiments [46]. Li et al. proposed an intelligent fault diagnosis method based on vibration signal gray image (GI), depth InfoMax (DIM) self-supervised learning (SSL) method, and convolution neural network (CNN) [47]. This method has good diagnostic performance and alleviates the over-fitting problem of model training caused by limited labeled samples. These methods have promoted the development of self-supervised learning, but there are also some shortcomings. Their diagnosis framework is complex, and the principle behind self-supervised learning requires further analysis. It can be found that only when the model learns sufficient useful features of vibration signals can it better complete the fault diagnosis task. It is difficult to obtain a large number of labeled samples, so the model feature extraction ability is insufficient.

Therefore, this paper conducts further research on self-supervised learning and uses a large number of unmarked samples to optimize the model, so that the model has better fault diagnosis ability. The sparse filtering algorithm is easy to implement and must adjust only one parameter (the number of features); thus, this paper chooses to study self-supervised learning based on the sparse filtering model. In addition, three fault diagnosis methods are selected and compared with the proposed self-supervised learning to verify the excellent performance of the proposed method. Zhang et al. [5] proposed a sparse filter extraction method based on time–frequency features, which can be used for diagnosis under different loads. Experimental results show that this method has good fault diagnosis ability and classification performance. Xue et al. [24] proposed a rolling bearing fault diagnosis method based on an entropy feature and a stacked sparse autoencoder. The experimental results show that the proposed method has higher fault diagnosis accuracy. Zhang et al. [48] proposed a generalized Sparse Representation based Classification (SRC) algorithm for open set recognition in which not all classes presented during testing are known during training. The effectiveness of the proposed method is verified by several examples, and the performance is obviously better than many open set recognition algorithms. It can be seen that sparse filtering (SF), Stacked Autoencoder (SAE), and Sparse Representation Classification Algorithm (SRC) have good fault diagnosis performance and can be used as a comparison method to compare with the proposed method.

## 3. Proposed Method

### 3.1. Methodology

GSLSF aims to improve the fault diagnosis performance with minimal training data and the generalization ability of the model. Based on this problem, an intelligent fault diagnosis method integrating self-supervised learning and sparse filtering is proposed. The proposed method can learn useful information from many unlabeled data and transfer it to the downstream tasks of model training, thereby improving the model classification accuracy and generalization ability. The proposed method is divided into two learning stages, namely model pre-training based on self-supervised learning and sparse filtering model training and classification based on knowledge transfer. Thus, the pretext tasks and the target tasks are established. Firstly, a data collector is used to collect a large number of unlabeled samples. Then, a self-supervised learning pretext task and pseudo-labels are designed based on the representation of samples of fault and operating condition information. In addition, the knowledge transfer mechanism is used to apply the useful features learned from the pre-training to the target task. Finally, CWRU dataset and rolling bearing experiments are used to verify the effectiveness of the GSLSF method. Figure 1 depicts the framework for the proposed self-supervised method GSLSF.

### 3.2. Two-Stage Learning Task

In addition, the knowledge transfer mechanism is used to apply the useful features learned from the pre-training to the target task. Figure 1 depicts the framework for the proposed self-supervised method GSLSF. In this section, the creation process of the two-stage learning task is described in detail. In the first learning stage, given an original dataset $D_{1}={\left\{X_{i}\right\}}_{i=1}^{N}$, a new dataset $D_{s}={\left(Y_{i}^{l},P_{i}^{l}\right)}_{i=1}^{N}$ is generated through multiple signal transformation types. This pseudo-label is automatically generated without manual annotation. Defining F(·) as a function of the signal transformation type yields $Y_{i}^{l}=F\left(X^{i}\right)$. *l* is an integer, and l∈(1,2,…,S), S is the total number of classes of signal transitions. Then, the generated data is taken as the training set, and the sparse filtering model is applied for pre-training to obtain the weight matrix W. The specific training process is as follows. First of all, the input dimension is set as $N_{in}$, and the output dimension is set as $N_{out}$. The input sample $D_{S}$ is divided into $N_{s}$ segments in a random way. This purpose of this step is to obtain random segments by overlapping. A new segment set ${\left\{S^{j}\right\}}_{j=1}^{N_{s}}$ consists of these random segments, where $S^{j}\in {ℜ}^{N_{in}\times 1}$ is the jth segment containing $N_{in}$ data points. The segment set ${\left\{S^{j}\right\}}_{j=1}^{N_{s}}$ is rewritten as a matrix form $S^{j}\in {ℜ}^{N_{in}\times N_{s}}$, and the sample set ${\left\{X^{i}\right\}}_{i=1}^{m}$ is rewritten as $x\in {ℜ}^{N_{in}\times M}$. The linear expression of training matrix and weight matrix is used to obtain local features.
(1)$$f_{j}^{i}=Wx$$
where $W\in {ℜ}^{N_{out}\times N_{in}}$ is the weight matrix, and $f_{j}^{i}$ is the jth feature of the ith segment. Then the average method is used to obtain the learned features.
(2)$$F^{i}=\frac{1}{N_{s}}\sum_{n=1}^{N_{s}}f_{n}^{i}$$

In the second learning stage, the knowledge transfer method is adopted. Transfer learning is one of the machine learning methods, and it refers to the transfer of the feature parameters learned by pre-training in the source domain to the target domain [34]. In this stage, the label set $D_{{}_{T}}={\left(X^{i},G^{i}\right)}_{i=1}^{N}$ is redefined, where $X^{i}$ is the raw vibration data and $G^{i}$ is the real fault label, and the overall class of health condition is G=1,2,3,4. As can be seen in our proposed framework, transferring the parameters of the first learning stage to the target task is a key operation to improve performance. Therefore, we freeze the weight matrix W obtained by pre-training in the first learning stage as the initialization weight matrix of the target task and perform the fault diagnosis task (target task). Finally, various fault types are identified using the softmax regression.

### 3.3. Pretext Task Design

To ease the workload of annotating large amounts of data, we generally set up a pretext task for which the pseudo-labels are automatically generated based on the inherent characteristics of the data. At present, researchers have invented many pretext tasks and have successfully applied them to self-supervised learning with good performance, such as foreground object segmentation [49], image completion [39], image colorization [41], etc. Only when an effective pretext task is established can the model learn effective features of the data. Therefore, there are many categories of signal transformation, and there is no fixed standard, as long as the pretext task that satisfies the self-supervised learning is established according to the actual situation. These pretext tasks share two characteristics. Firstly (1), the model must learn useful features to solve the pretext task. Secondly (2), the creation of pseudo-labels is automatically generated based on a certain characteristic of the signal. In this paper, a new signal transformation pretext task is proposed based on the transformation characteristics of signals. Six signal transformation types are adopted, namely normal, value increase, value decrease, random zeroing, adding white noise, and disordered splicing. We argue that if the signal transition types in the pretext task can be identified by the model, then the model after knowledge transfer can identify the types of failures in the target task, thereby improving the robustness and generalization ability of the model. The following is an introduction to the six types of signal transformations.

Normal. Given an original vibration signal $X\left(i\right)=\left[x_{1},x_{2},\ldots ,x_{i}\right]$, where *i* represents the length of the signal, without any signal transformation, so that Y(i)=X(i).

Value increase. A signal X(i) and a signal conversion vector $B\left(i\right)=\left[b_{1},b_{2},\ldots ,b_{i}\right]$ are given, and the value of the original vibration signal is amplified by linear multiplication to obtain the vibration signal $Y\left(i\right)=X\left(i\right)B\left(i\right)=\left[x_{1}b_{1},x_{2}b_{2},\ldots ,x_{i}b_{i}\right]$ after the value is increased.

Value decrease. Given a signal X(i) and a signal transformation vector $B\left(i\right)=\left[b_{1},b_{2},\ldots ,b_{i}\right]$, the signal X(i) divided by the corresponding element of B(i) gives the transformed $Y\left(i\right)=X\left(i\right)/B\left(i\right)=\left[x_{1}/b_{1},x_{2}/b_{2},\ldots ,x_{i}/b_{i}\right]$.

Add Gaussian white noise. Gaussian white noise is added to the vibration signal X(i), which is a random signal with a constant power spectral density. We obtain the signal converted from the signal Y(i).

Disordered splicing. The signal X(i) is cut into *m* segments, namely $X\left(i\right)=\left[X_{1}\left(i\right),X_{2}\left(i\right),X_{3}\left(i\right),\ldots X_{m}\left(i\right)\right]$. In addition, these fragments are randomly combined into a new signal set Y(i).

Random zeroing. Given a signal X(i) and a signal transformation vector $B\left(i\right)=\left[b_{1},b_{2},\ldots ,b_{i}\right]$, where 25% of the elements are 0, and 75% of the elements are 1. Use multiplication to obtain the converted vibration signal $Y\left(i\right)=X\left(i\right)B\left(i\right)=\left[x_{1}b_{1},x_{2}b_{2},\ldots ,x_{i}b_{i}\right]$.

Pretext tasks are established based on the above six signal transformation types. During training of the surrogate task, the signal transformation type is randomly selected to transform the original vibration signal, and its corresponding pseudo-label P=[1,2,…,6] is generated. The sparse filtering model is used to pre-train the pretext task, and the softmax regression is applied to distinguish different types of signal change. The model can thereby learn the meaningful information of the signal. The learned weight matrix is transferred to the target task to complete the fault diagnosis task. As can be seen from Figure 2, the samples processed by these six signal transformation methods are very similar, but there are great differences in nature. This requires the network to learn the essential characteristics of the signal, so as to accurately classify the signal type.

## 4. Experimental Validation

### 4.1. Case 1: Bearing Fault Diagnosis Based on Case Western Reserve University Dataset

#### 4.1.1. Dataset Description

In this section, the proposed self-supervised learning approach GSLSF is experimentally validated using rolling bearing data from Case Western Reserve University [50]. The instruments and equipment of the test bench have acceleration sensors, rolling bearings, and loading motors. The sampling frequency of 12 kHz is used to collect vibration signals at the motor drive end of the test bench. Bearings have four different health states: normal (Nor), inner ring fault (IR), outer ring fault (OR), and ball fault (B). Each fault type corresponds to three different severities, and the fault sizes are 0.18 mm, 0.36 mm, and 0.53 mm, respectively. The vibration signals of the bearing under loads 0, 1, 2, and 3 hp were collected. The rotational speeds of the four loads are 1797 r/min, 1772 r/min, 1750r/min, and 1730 r/min, respectively. For the convenience of representation, the samples composed of signals under four different loads are referred to as A, B, C, and D. The faults with the same fault location but different severity are regarded as one kind of operating condition; thus, there are 10 failure types in total. There are 100 samples under each load, and each sample includes 1200 points; thus, the bearing dataset has a total of 4000 samples. The details of this dataset are shown in Table 1.

#### 4.1.2. Analysis of Parameter Sensitivity

To verify the superiority of GSLSF, the analysis of sensitive parameters is carried out. The difference in sensitive parameter values can affect the fault diagnosis accuracy and robustness; thus, we make optimal choices for the input dimension, output dimension, and the number of segments in the target task. In the following experiments, we use A, B, C, and D to denote the four loading conditions of 0 Hp, 1 Hp, 2 Hp, and 3 Hp, respectively. The samples under A are selected to calculate the best sensitive parameters. The self-supervised learning method is proposed by randomly selecting 10% samples for training, that is, 12,000 sample points are used for training, and the rest are tested. In order to avoid the influence of randomness, a total of 10 experiments is carried out for each experiment. The average accuracy and average computation time of the experiments are obtained from 10 trials. In this experiment, the weight decay λ is set to 1 × 10^−5^. By default, the input dimension $N_{in}$ is set to 40, the output dimension $N_{out}$ is set to 80, and the number of segments $N_{s}$ in both the pre-training task and the target task is set is 50.

Firstly, the optimal number of segments for the target task is selected. Input dimension $N_{in}$ and output dimension $N_{out}$ are set to 40 and 80, respectively. Figure 3a is the accuracy rate corresponding to a different number of segments. As can be seen from Figure 3a, the size of the number of segments has a great influence on diagnostic accuracy. As can be analyzed from Figure 3a, the accuracy rate becomes higher, and the average calculation time increases linearly. When the number of segments $N_{s}$ > 10, the average accuracy rate is greater than 95%, and the proposed self-supervised method can distinguish the ten health conditions of the bearing dataset with high precision. When the number of segments $N_{s}$ > 120, the average accuracy basically does not fluctuate, but the average calculation time becomes longer. Through further analysis, we found that when $N_{s}$ = 120, the calculation effect is significantly better than other segment numbers. Therefore, the number of segments $N_{s}$ is chosen to be 120 in this experiment.

Next, select the input dimension $N_{in}$, where $N_{out}$ = 80 and $N_{s}$ = 50. Figure 3b shows the fault diagnosis results of different input dimensions. When $N_{in}$ > 40, the fault diagnosis accuracy gradually decreases, and the calculation time increases approximately proportionally. When $N_{in}$ = 40, the fault diagnosis accuracy is the highest, and the standard deviation is also small, showing better performance. Therefore, we choose $N_{in}$ = 40 as the best input dimension for this experiment.

Finally, we obtain the optimal parameter for the output dimension $N_{out}$, where $N_{in}$ = 40 and $N_{s}$ = 50. The influence of different output dimensions on the diagnostic results is depicted in Figure 3c. When the output dimension is $N_{out}$ > 40, the average diagnosis rate is greater than 93%, which indicates that the method can accurately classify the faults. The computation time grows linearly as the output dimension increases. When $N_{out}$ = 80, the fault diagnosis accuracy is the highest, the standard deviation is smaller, and the performance is superior to other output dimension values. Therefore, we choose a value of 80 for the output dimension $N_{out}$.

#### 4.1.3. Results and Analysis

Comparison approaches: three machine learning methods which have achieved remarkable success in intelligent fault diagnosis are compared with the proposed method to verify the effectiveness of the proposed method.
(1)Sparse filtering (SF) [5], which is a novel shallow machine learning approach, and the model is easy to achieve convergence.(2)Stacked Autoencoder (SAE) [24] is an advanced deep network architecture which can extract deep features and is widely used in fault diagnosis.(3)Sparse Representation Classification Algorithm (SRC) [48] uses class reconstruction errors for classification.


In this part, we employ three methods to compare the performance of the proposed method. It is worth noting that the bearing datasets used by the three methods are all in Table 1. Each experiment is conducted 10 times in order to reduce the randomness that interferes with the experiment. Finally, the average accuracy is calculated. Figure 4 and Table 2 show the average diagnostic accuracy of the three methods for the entire dataset and the dataset under each loading, respectively. The input dimension $N_{in}$ and output dimension $N_{out}$ of the proposed method are set to 40 and 80, respectively. The number of pre-training segments $N_{s}$ is 50, and the number of segments $N_{s}$ in the target task is set to 120. In the pre-training task, we randomly draw eight samples, including Nor1; Nor2; Nor3; Nor4; B11; IR12; OR11; and OR13. Then, six kinds of signal transformations are performed, the sparse filtering is trained to obtain the weight matrix, and the parameters are moved to the target task through knowledge transfer. In the target task, 10% of the samples are used for training and 90% samples are tested. Finally, the fault types of the test samples are classified by softmax regression. In Table 2, the proposed self-supervised method has an average accuracy of about 97.26% in most fault diagnosis tasks, and the fault diagnosis accuracy and generalization ability are significantly better than the comparison methods, which verifies the effectiveness and robustness of the proposed method. It can be found that the diagnostic accuracy of the self-supervised method in the total dataset is higher than that of the datasets under the four working conditions of A, B, C, and D. Because the self-supervised method requires minimal labeled data for pre-training, and the labeled data in the total dataset are more, the model performance is better. In addition, it can be seen that the larger the sample size, the smaller the standard deviation, and the stronger the model generalization ability. Generally, calculation efficiency is an important index for evaluating the effectiveness of the model. It can be seen from Figure 5 that the SRC method has the longest running time, and the calculation efficiency is poor. The SAE method has a short running time, and the proposed method has a running time close to that of the SF method. In general, the proposed method has good computational efficiency.

The three methods are compared with the proposed method to prove the superiority of the proposed method. The first is the traditional sparse filtering feature extraction method, which we call SF for short. The parameter settings are as follows: $N_{in}$ = 40, $N_{out}$ = 80, $N_{s}$ = 50, λ=1E−5. In total, 10% of the samples are used in training, and the rest are used to test the learning effect of the model and calculate the diagnostic accuracy under each load. As can be seen from Table 2, the diagnostic precision of this method is around 93.67~97.41%, and the average accuracy is 96.06%, which is 1.1% lower than that of the proposed method. In addition, the standard deviation of this method is larger than that of the proposed method, and the model performance is more unstable. Therefore, compared with the proposed method, this method has lower diagnostic accuracy and poorer stability.

The second method is the Stacked Autoencoder, call SAE. The parameter is set as follows. The number of hidden neurons is chosen as 100, and the number of iterations is 100. Batch size and learning rate are set to 50 and 3.5, respectively. Of the samples, 10% are used for training, and 90% are used for testing. Finally, softmax regression is used to classify the bearing health. In Table 2, the diagnostic accuracy of this method is about 90.41~98.29%, and the average accuracy rate is 93.61%, which is about 3.65% lower than that of the proposed method, and the stability is poor. When there are minimal training data, the diagnostic effect of the method has obvious changes; thus, the method is more dependent on the amount of training data. The third method is the Sparse Representation Classification Algorithm, also known as SRC. Of the samples, 50% are used for training, and 50% are used for testing. As can be seen from Table 2, the accuracy rate of this method is 76.16~90.16%, which is worse than the diagnostic performance of the proposed method. It can be seen from the fault accuracy rate under various working conditions that the robustness of this method is poor. To sum up, our proposed method is of great help for bearing fault diagnosis, and the method can play a better performance under limited labeled samples.

The confusion matrix is applied to further show the fault classification results clearly, as shown in Figure 6. Figure 6 shows the diagnostic results of the proposed method and of SF and SAE under A and B conditions, respectively. It is worth noting that there are 90 samples for each failure type, and all are unlabeled data. It can be concluded from Figure 6a that in case A, there are only nine misclassifications in the proposed method. Among them, fault 3 is wrongly classified as one fault 7, one fault 4, and one fault 6. Fault 10 is wrongly classified as three faults 2, one fault 7, one fault 9, and three faults 2, respectively. In addition, the sparse filtering method has a total of 20 samples misclassified in case A, which is less effective than the proposed method. From Figure 6b, we can see that the classification effect of SAE is worse than the proposed method. In the proposed method, a total of 24 samples are misclassified, and the misclassification results are mainly concentrated on label 10, and 22 samples are misclassified as label 6. However, a total of 50 samples are misclassified by SAE, the misclassified results are scattered, and the overall classification results are poor. In conclusion, the proposed method has better fault classification performance.

### 4.2. Case 2: Fault Diagnosis for Rolling Bearings of Special Test Bench

#### 4.2.1. Dataset Description

In order to verify the superiority of the self-supervised sparse filtering method, as shown in Figure 7, an experimental platform is employed to simulate bearing failure. The platform includes the electric machinery, bearing housing, gear box, and brakes. The vibration data of the bearing base is collected by an LMS data acquisition instrument with a vibration sensor, and the sensor is placed on the side of the bearing housing. The sampling frequency is 25.6 kHz. The three bearing types are N205EU cylindrical roller bearings, as shown in Figure 8. As shown in Table 3, the bearing has four different health conditions: normal (NC), outer ring fault (OF), inner ring fault (IF), and ball fault (RF). At the same time, each health condition is damaged by three different degrees of severity: 0.18 mm, 0.36 mm, and 0.54 mm. Thus, there are ten health states of the bearing; each health state includes 100 samples, and one sample includes 2560 data points.

#### 4.2.2. Analysis of Parameter Sensitivity

Firstly, the best parameters for this experiment are set by selecting the parameters in Section 4.1.2. As shown in Figure 9, the input dimension $N_{in}$, the output dimension $N_{out}$, and the segment number $N_{s}$ in the target task are the optimal parameter selection. Figure 9a shows that the average precision increases as the number of segments increases. When $N_{s}$ > 30, the accuracy rate is above 90%, indicating that the method can more accurately classify bearing faults. Considering the average accuracy and computation time, $N_{s}$ = 120 is chosen as the optimal number of segments for this experiment. Through the same method, the input dimension is $N_{in}$, and the output dimension is $N_{out}$ again. From Figure 9b,c, it can be concluded that the input dimension $N_{in}$ and the output dimension $N_{out}$ are selected as 40 and 80, respectively.

#### 4.2.3. Results and Analysis

This section adopts the same method as Section 4.1.3, and the advantage of the proposed approach is proved. The data used in this experiment are listed in Table 3. The diagnostic precision of the four approaches is shown in Figure 10 and Table 4. Each experiment is run 10 times in order to reduce the randomness that interferes with the experiment. As can be seen from Table 4, the diagnostic precision of the proposed approach is about 95.27~97.87%, the average precision is 97.02%, the standard deviation is small, and the robustness is good. From Figure 10 and Table 4, it can be observed that the diagnostic accuracy of the proposed approach is better than that of SF, SAE, and SRC under various working conditions, and the standard deviation is smaller than SF, SAE, and SRC in most working conditions, indicating that the robustness performs well. Under all conditions, the diagnostic accuracy of the proposed approach is 0.85% higher than that of SF, and 1.06% higher than that of SAE, which proves the advantage of the proposed approach in fault feature classification. In addition, it can be found that the SAE method has a very poor diagnostic effect under the four working conditions of A, B, C, and D, and the standard deviation is large. Meanwhile, the diagnostic accuracy of SRC was 76.14–89% in each working condition, the diagnostic performance was poor, and the standard deviation was large. Through analysis, this method is relatively dependent on the amount of training data, and it requires a large amount of data to learn reliable features. When the training data increases, the effect is significantly improved. This article does not discuss this topic in depth.

To further prove the advantages of the method, we apply t-SNE to visually classify bearing health conditions. Figure 11 is the fault classification diagram of GSLSF, and the traditional sparse filtering method under working conditions A and B. From Figure 11a,b, it can be seen that most of the features are correctly classified, and only a small number of rolling element faults are confused. From Figure 11c, it can be seen that the proposed method correctly classifies health status. However, there are some problems in the classification of the traditional sparse filtering method in Figure 11d, and a few inner-circle faults are confused. In general, the classification performance of the proposed approach is clearly better than that of SF.

## 5. Conclusions

In this paper, we focus on using minimal labeled data to enhance the feature extraction ability and generalization ability of the model and propose an intelligent fault diagnosis method that combines self-supervised learning and a sparse filtering model. Because the vibration signal has the characteristics of time sequence and continuity, we establish a pretext task and pseudo-label for self-supervised learning based on the signal representation method. Under the condition of a small amount of labeled data, the vibration signal variation characteristics are learned by pre-training, and then the learned weight matrix is applied to the target task through the knowledge transfer mechanism, so as to accurately classify the fault. This approach can facilitate the learning of valuable information from many unlabeled vibration signals. It renders the network more efficient in extracting signal features. This approach has been verified on both the CWRU dataset and the bearing data collected in the experiment, and the method has good performance in fault diagnosis. The experimental results show that GSLSF has high precision and good robustness and generalization ability. Therefore, the method can facilitate learning of sample features from the few labeled training data and improve the fault diagnosis ability of the model.

However, this method also has some disadvantages. Because the self-supervised method requires a pre-training process, which requires the network to learn the change of signal category of the pretext task, the total computing time will increase slightly, and the computational efficiency is lower than that of the traditional sparse filtering method. Second, the training of self-supervised methods requires a certain amount of labeled data to train the model to optimize the decision boundary.

In future work, we can improve the pretext task and create different pretext tasks to classify faults. Secondly, the model pre-trained by the self-supervised learning method can be applied to a variety of fields, and this paper only analyzes the task of fault diagnosis. In the future, we can research how this approach performs in other areas.

## Figures and Tables

**Figure 1 sensors-23-01858-f001:**
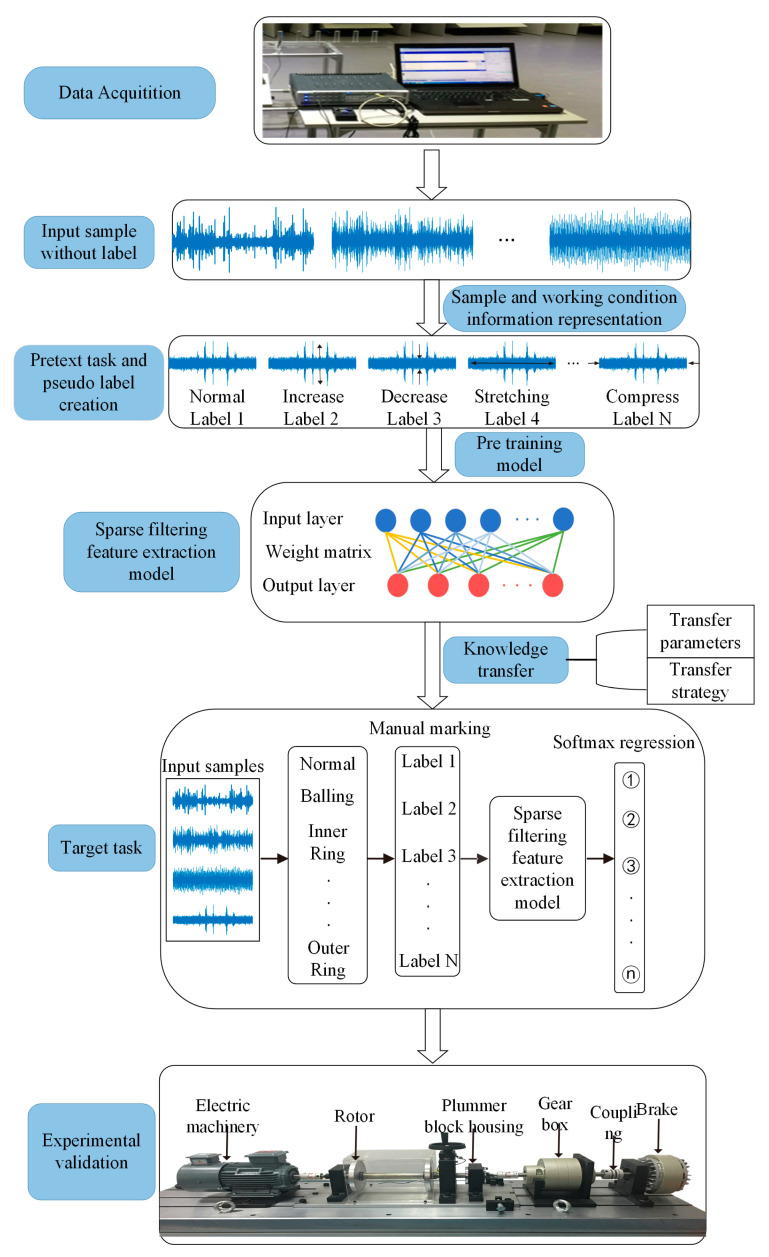
The details of the proposed self-supervised learning framework.

**Figure 2 sensors-23-01858-f002:**
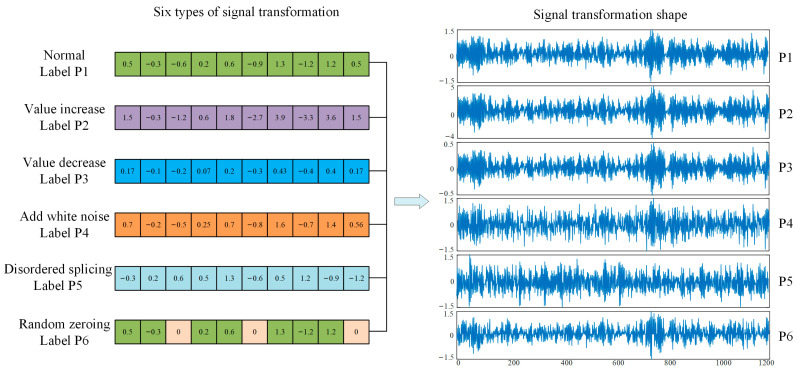
Shapes of six signal transformation methods.

**Figure 3 sensors-23-01858-f003:**
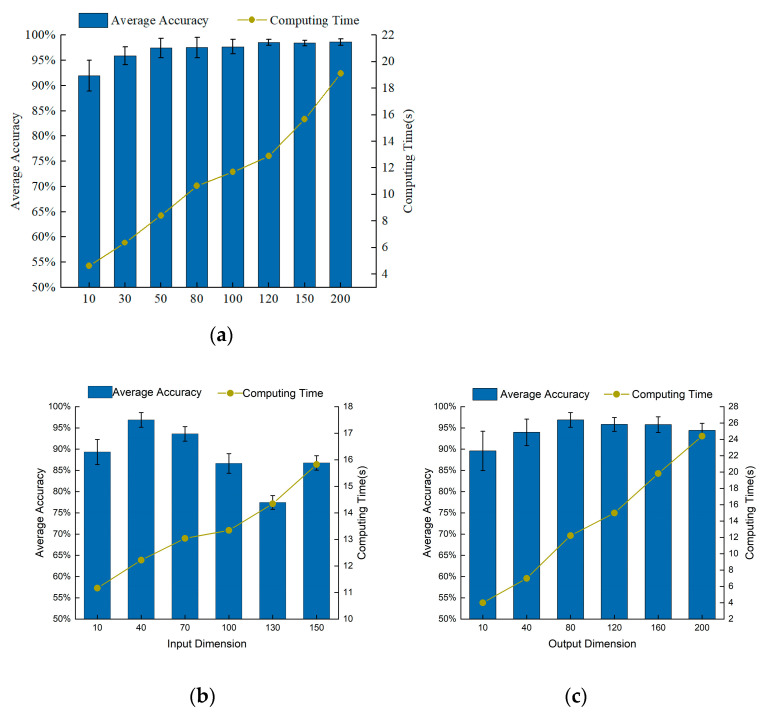
Influence of various parameters on diagnostic performance of GSLSF. (**a**) Number of segments. (**b**) Input dimension. (**c**) Output dimension.

**Figure 4 sensors-23-01858-f004:**
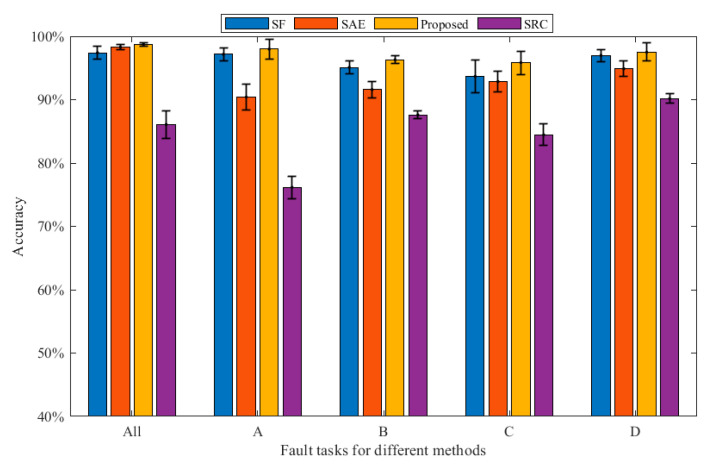
Bearing diagnosis results of the three methods under various working conditions.

**Figure 5 sensors-23-01858-f005:**
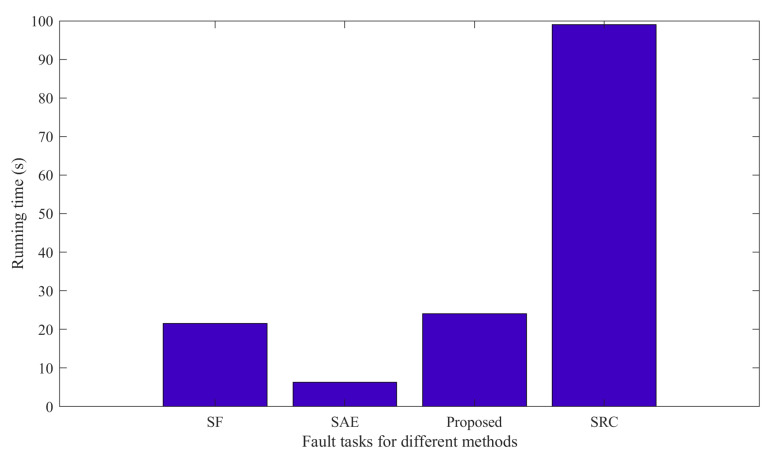
Running time for different methods.

**Figure 6 sensors-23-01858-f006:**
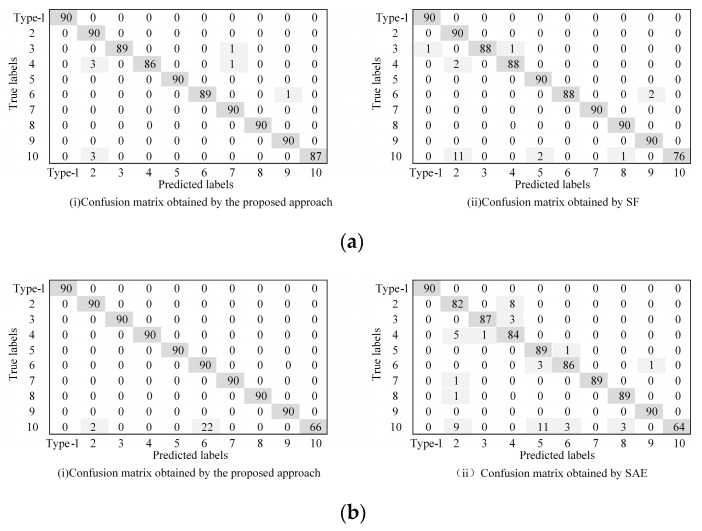
Comparison of the confusion matrixes obtained by different approaches. (**a**) Confusion matrix in case A. (**b**) Confusion matrix in case B.

**Figure 7 sensors-23-01858-f007:**
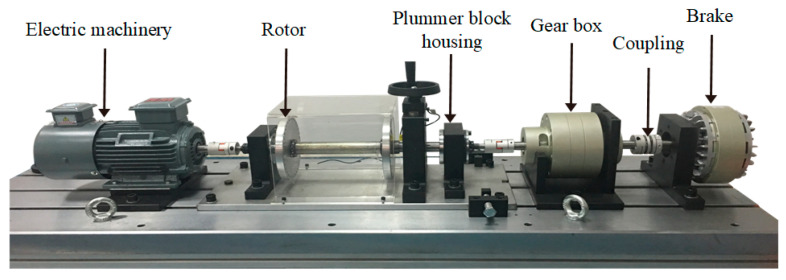
Ball bearing data testing platform.

**Figure 8 sensors-23-01858-f008:**
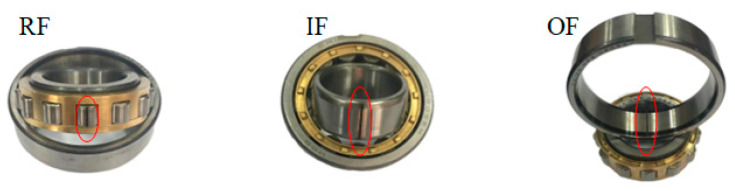
Structure display of rolling bearing.

**Figure 9 sensors-23-01858-f009:**
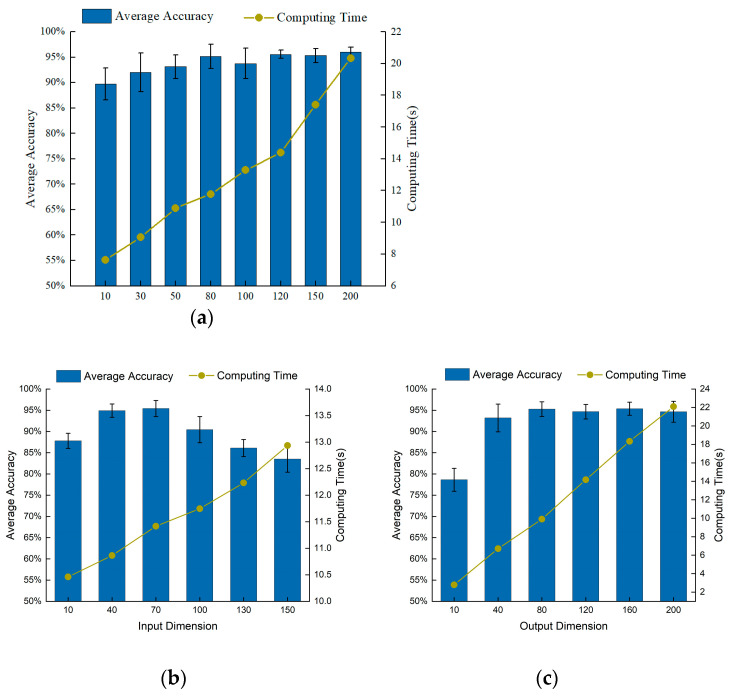
Influence of various parameters on diagnostic performance of GSLSF. (**a**) Number of segments. (**b**) Input dimension. (**c**) Output dimension.

**Figure 10 sensors-23-01858-f010:**
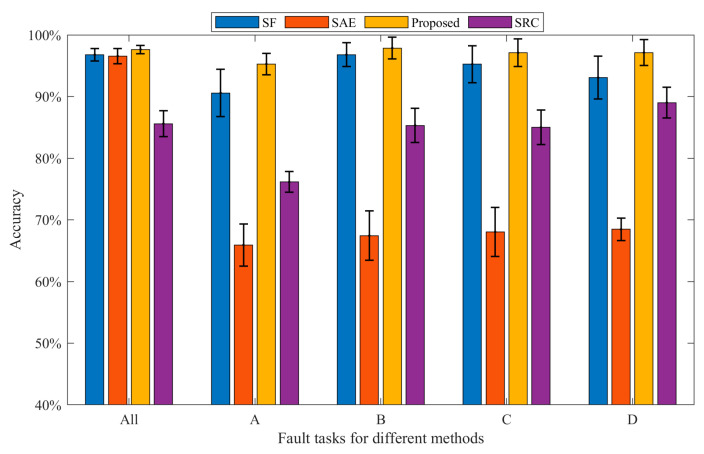
Bearing diagnosis results of the three methods under various working conditions.

**Figure 11 sensors-23-01858-f011:**
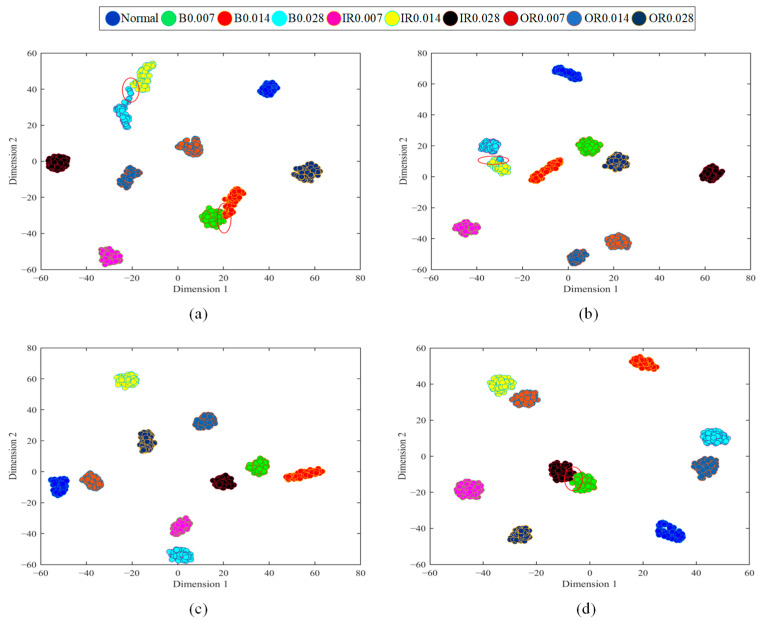
Visualization diagram of 2-D bearing fault features. (**a**) The proposed approach in A. (**b**) SF in A. (**c**) The proposed approach in D. (**d**) SF in D.

**Table 1 sensors-23-01858-t001:** Composition of the datasets.

Bearing Fault Pattern	Fault Size (mm)	Load (/hp)	Label	Abbreviation
Normal	/	0, 1, 2, 3	1	Nor
Inner race fault	0.18	0, 1, 2, 3	2	IR1
	0.36	0, 1, 2, 3	3	IR2
	0.53	0, 1, 2, 3	4	IR3
Ball fault	0.18	0, 1, 2, 3	5	B1
	0.36	0, 1, 2, 3	6	B2
	0.53	0, 1, 2, 3	7	B3
Outer race fault	0.18	0, 1, 2, 3	8	OR1
	0.36	0, 1, 2, 3	9	OR2
	0.53	0, 1, 2, 3	10	OR3

**Table 2 sensors-23-01858-t002:** The detailed diagnostic results (%) of the bearing data.

Methods	SF	SAE	SRC	Proposed
All	97.41 ± 1.04	98.29 ± 0.40	86.08 ± 2.16	98.72 ± 0.31
A	97.16 ± 1.06	90.41 ± 2.10	76.16 ± 1.76	97.98±1.58
B	95.09 ± 0.97	91.59 ± 1.33	87.6 ± 0.61	96.29 ± 0.60
C	93.67 ± 2.61	92.86 ± 1.61	84.44 ± 1.71	95.84 ± 1.83
D	96.98 ± 0.97	94.88 ± 1.21	90.16 ± 0.76	97.49 ± 1.43
Ave	96.06	93.61	84.88	97.26

**Table 3 sensors-23-01858-t003:** The description of the datasets.

Health Condition	Sampling Frequency	Fault Size (mm)	Label
NC	25.6 kHz	0	1
IF1		0.18	2
IF2		0.36	3
IF3		0.54	4
OF1		0.18	5
OF2		0.36	6
OF3		0.54	7
RF1		0.18	8
RF2		0.36	9
RF3		0.54	10

**Table 4 sensors-23-01858-t004:** The detailed diagnostic results (%) of the bearing data.

Methods	SF	SAE	SRC	Proposed
All	96.78 ± 0.99	96.57 ± 1.23	85.60 ± 2.1	97.63 ± 0.68
A	90.58 ± 3.82	65.92 ± 3.41	76.14 ± 1.68	95.27 ± 1.73
B	96.79 ± 1.94	67.43 ± 4.00	85.32 ± 2.77	97.87 ± 1.74
C	95.24 ± 2.99	68.04 ± 4.00	85.00 ± 2.8	97.11 ± 2.22
D	93.07 ± 3.46	68.46 ± 4.00	89.00 ± 2.5	97.13 ± 2.09
Ave	94.49	73.28	84.21	97.02

## Data Availability

The raw vibration data used in the paper can be download from https://engineering.case.edu/bearingdatacenter/12k-drive-end-bearing-fault-data (accessed on 10 September 2021).

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
