# Peer review of "A Novel Study on a Generalized Model Based on Self-Supervised Learning and Sparse Filtering for Intelligent Bearing Fault Diagnosis"

_sensors, 2023, doi:10.3390/s23041858_

Round 1

Reviewer 1 Report

Problem definition is not clear in introduction part, author should define problem properly in context of data source. structure of manuscript should modify, please add experimentation and data collection part in methods section, and dnt repeat the Result and analysis part, both cases should be mentioned under Result and discussion heading, Furthermore small chages should be:

Line 50-51, “It can be found that intelligent fault diagnosis research has achieved a lot of good 50 results” statement claim result, but theoretical proofs are not provided (add reference)

Figure 1:text should be rewrite, it not well readable.

Its better to not to use the first person singular as , we etc.. change the sentence to passive or past tens as have been etc.

Reviewer 2 Report

The paper presents an interesting subject, but the following aspects must be improved:

-sections 2.1 and 2.2 must be removed - they contain only theoretical information regarding some aspects that are not relevant for a research article; these theoretical aspects must be used (as it is) in the description of the proposed solution

-a section with related works must be added (including also obtained results) – relevant existing solutions correlated with the proposed method (with the scope of the method must be added), including also the obtained results (advantages and disadvantages of the existing solutions). Based on these results (proven disadvantages) show what are the benefits of the proposed solution

-the proposed method must be described in more details in order to prove the benefits of the proposed method / novelty of it – in comparison with existing results shown in section with related works

-methods SF and SAE must be presented first in the section with related works in order to show why these are selected for comparison (do they have the best results? or what are their existing benefits)

-in order to show performances of the proposed solution, evaluation must be more elaborated; comparison with more existing methods must be added; 

Reviewer 3 Report

Check that the style of writing is in the third person throughout. Don’t use ‘we’.

The authors mentioned that 10% of samples are used for training and 90% for testing. It isn't very clear. Usually, more % of acquired data is used for training. Additionally, results must be provided considering different holdout % and holdout validation approaches. You may refer to the article to understand the holdout validation approach. You may refer to the following article https://doi.org/10.1115/1.4051696

Results for classification using a blind dataset (No labels) should be provided after training the model. Apply the trained model to classify blind datasets. You may refer to “Figure 8: Framework for classification of blind data” from the following article. DOI is https://doi.org/10.36001/ijphm.2020.v11i2.2929

Was the data normalized/ standardized?

Was the algorithm trained using standard hyperparameters, or were they altered?

Comment on computational complexity in the training of algorithm in terms of its deployment in real-time.

Hyperparameters of the designed network must be included in a tabular form.

A comparison of deep learning and machine learning algorithms should be provided by referring to the following articles.

·       Deep learning algorithms for tool condition monitoring in milling: A review

·       Data Driven Cutting Tool Fault Diagnosis System Using Machine Learning Approach: A Review

How to deal with the diversity between the data distributions of present and future moments? ML-based algorithms can only resolve classification issues within the same data distributions. What would be the key steps of generalizing to unknown moments in predicting various parameters?

How to deal with the misclassification of a normal condition as a faulty condition depending on the degree of fault?

How to deal with the misclassification of faulty conditions as normal conditions depending on the degree of fault (type II error)? If the model is deployed in real-time and such a situation arises, how will you identify that the blade is in the failure zone and showcased as normal by your system?

How to ensure the robustness of the model in a high noisy environment?

Tree-based algorithms exhibit significant performance and can explain the reasoning behind decisions by ML model. Refer to the paper, and you may suggest white box modeling using tree-based algorithms in the future scope. ‘A white-box SVM framework and its swarm-based optimization for supervision of toothed milling cutter through characterization of spindle vibrations’

I am trying to give future directions by suggesting some papers cited here. You may refer them if you find them interesting.

Round 2

Reviewer 1 Report

Accept

Author Response

Thank you again for your review, these valuable and professional suggestions help us improve our work dramatically.

Reviewer 2 Report

Some of my comments were addressed. There is one aspect that must be considered:

-related work section must contain also results in order to prove the selection of the methods SF, SAE and SRC for comparison with the proposed method (the comparison must be made with existing methods with best performances).

Reviewer 3 Report

The authors tried to address my comments.

Author Response

(The authors gave the same response as above.)

Round 3

Reviewer 2 Report

Section 4.1.3. Results and analysis contains general information about SF, SAE and SRC but it is needed to add also quantitative results (metrics) in order to prove selection of these methods for comparing the proposed method with other existing ones (show clearly what are existing performances).
